# Multidisciplinary Assessment and Individualized Nutritional Management of Dysphagia in Older Outpatients

**DOI:** 10.3390/nu15051103

**Published:** 2023-02-22

**Authors:** Nikolina Jukic Peladic, Paolo Orlandoni, Mirko Di Rosa, Giulia Giulioni, Laura Bartoloni, Claudia Venturini

**Affiliations:** 1Vivisol srl., Clinical Nutrition Unit, National Institute of Health and Science on Aging, IRCCS INRCA Ancona, Via della Montagnola 81, 60127 Ancona, Italy; 2Clinical Nutrition Unit, National Institute of Health and Science on Aging IRCCS INRCA Ancona, Via della Montagnola 81, 60127 Ancona, Italy; 3Unit of Geriatric Pharmacoepidemiology and Biostatistics, National Institute of Health and Science on Aging, IRCCS INRCA Ancona, Via Santa Margherita 5, 60124 Ancona, Italy

**Keywords:** dysphagia, geriatrics, outpatient, nutrition therapy, texture-modified diet, GLIM, IDDSI

## Abstract

Introduction: The evidence on the efficacy of nutrition therapy to prevent complications of dysphagia is based on observational studies that used different tools for nutritional and dysphagia assessment, and different scales for the definition of diet textures, rendering their results incomparable and the knowledge on dysphagia management inconclusive. Methods: This retrospective observational study was performed in 267 older outpatients who were assessed for dysphagia and nutritional status by a multidisciplinary team at the Clinical Nutrition Unit of IRCCS INRCA geriatric research hospital (Ancona, Italy) from 2018 to 2021. GUSS test and ASHA-NOMS measurement systems were used for dysphagia assessment, GLIM criteria for the assessment of nutritional status, and the IDDSI framework to describe the texture-modified diets. Descriptive statistics were used to summarize the characteristics of the subjects evaluated. Sociodemographic, functional and clinical parameters were compared between patients with and without BMI improvement overtime by an unpaired Student’s *t* test, Mann–Whitney U test or Chi square test, as appropriate. Results: Dysphagia was diagnosed in more than 96.0% of subjects; 22.1% (n = 59) of dysphagic subjects were also malnourished. Dysphagia was treated exclusively by nutrition therapy, prevalently by individualized texture-modified diets (77.4%). For the classification of diet texture, the IDDSI framework was used. The follow-up visit was attended by 63.7% (n = 102) of subjects. Aspiration pneumonia was registered only in one patient (less than 1%), and BMI improved in 13 of 19 malnourished subjects (68.4%). The improvement of nutritional status was primarily reached in subjects whose energy intake was increased and texture of solids modified, in younger subjects, and in those taking less drugs and not reporting any weight loss before the first assessment. Conclusions: The nutritional management of dysphagia must guarantee both an adequate consistency and energy–protein intake. Evaluations and outcomes should be described with universal scales, in order to allow for comparison between studies and contribute to the collection of a critical mass of evidence on the efficacy of texture-modified diets in the management of dysphagia and its complications.

## 1. Introduction

Ageing is responsible for changes in the swallowing process that, in some cases, may compromise the efficacy and safety of swallowing and cause the medical condition known as dysphagia [1,2]. Swallowing disorders are categorized according to the swallowing phase that is affected. Oropharyngeal dysphagia (OD) is predominant in older subjects [3,4,5] and particularly frequent in those with neurologic disorders and in nursing home residents (NHRs). Among NHRs, the prevalence of dysphagia ranges between 8.1% and 93.0% [6,7,8], depending on the diagnostic tools and instruments used in the studies.

To prevent the onset of complications of dysphagia–aspiration pneumonia, malnutrition and dehydration, nutritional intervention is mostly adopted, with the texture modification of foods and liquids as the most frequent practice. The initiation of artificial nutrition (AN) is only sporadic [9,10,11]. Yet, for different reasons, the evidence on the adequacy and efficacy of texture modification for the management of malnutrition is scarce [12]. Some studies report the results of clinical practices where texture modification is not necessarily preceded by an accurate assessment of the protein–calorie needs of subjects with dysphagia, who are simply supposed to be malnourished. In some other studies, the nutritional assessment was carried out with unreliable tools, also due to the lack of a gold standard for its evaluation. In addition, due to the lack of an internationally accepted terminology, very different scales and levels were used for the classification of textures prescribed, making the results of different studies incomparable [13].

Scientific societies have recently proposed some tools that can help to overcome these gaps. In 2019, the major global clinical nutrition societies proposed the Global Leadership Initiative on Malnutrition (GLIM) criteria for the diagnosis of malnutrition; they have a high diagnostic accuracy in identifying patients with malnutrition and the potential to be used as a gold standard [14,15]. In 2013, the International Dysphagia Standardization Initiative (IDDSI) was founded to develop a standardized terminology for naming and classifying texture-modified food and liquids in all settings and cultures [16].

The goal of this study was to describe the characteristics and the nutrition therapies of older outpatients whose swallowing difficulties and nutritional status were assessed in a geriatric research hospital by recently proposed tools and scales. The detailed description of the use of standardized scales and tools could contribute significantly to gathering the critical mass of knowledge that is necessary to evaluate the effectiveness of texture-modified diets in preventing aspiration pneumonia and treating malnutrition, and to overcoming the knowledge gaps that prevent an efficient and evidence-based management of dysphagia.

## 2. Materials and Methods

INRCA’s ethics committee approved the study protocol in compliance with Italian national rules and standards for ethical research conduct (approval n.22034 from 19/01/23). All older outpatients without a previous diagnosis of dysphagia who signed their informed consent and underwent the assessment of dysphagia and nutritional status at the Clinical Nutrition Unit of the National Institute of Health and Science on Aging (IRCCS) INRCA of Ancona, in the period from 2018 to 2021, were included. Outpatient visits were performed according to a regular protocol of the Clinical Nutrition Unit by a multi-professional team, formed of a physician nutrition specialist, dietitians and speech therapists. Standard demographic data on age, gender and living conditions were collected during interviews with patients and/or their caregivers at the first visit. Detailed data on comorbidities and medications were collected from the patients’ documentation. Values of serum albumin, serum prealbumin and C reactive protein (CRP) were collected. Two days prior to the visit, the patients were invited to fill in food diaries containing information on food and drinks consumed, as well as their texture, and sensations and difficulties experienced when swallowing.

### 2.1. Dysphagia Assessment

The speech therapist performed differential diagnoses of dysphagia. Information on the main dysphagia-related sensations and symptoms were gathered during the interviews with patients and from food diaries. The speech therapist assessed for each patient the ability to control the trunk, to cooperate during the visit, communicative deficits, awareness and oral health. For dysphagia screening, the Gugging Swallowing Screen (GUSS) was used, which allows for the assessment of dysphagia severity and the risk of aspiration for both fluid and non-fluid foods [17,18]. The GUSS test begins with the assessment of vigilance, voluntary cough and/or throat clearing and saliva swallowing (swallowing, drooling, voice change); it then proceeds to test the swallowing of semisolid, fluid and solid textures. Four levels of dysphagia severity, accompanied by different diet recommendations, are possible: severe dysphagia and high aspiration risk (0–9), moderate dysphagia and moderate risk of aspiration (10–14), mild dysphagia with mild aspiration (16–19) and normal swallowing (20). During the assessment of dysphagia, the American Speech–Language–Hearing Association (ASHA) Functional Communication Measure swallowing subscale was also used to rate dysphagia severity based on the patient’s ability to meet nutritional needs and independence with compensatory strategies. This 7-point scale ranges from Level 1, indicating that the individual is not able to swallow anything safely by mouth and AN is necessary, to Level 7, indicating the absence of any limitation to safe swallowing [19].

### 2.2. Nutritional Assessment and Nutrition Therapy

Dietitians and nutritionists used the five-step Malnutrition Universal Screening Tool (MUST) for malnutrition screening [20]. Patients’ weight and height were measured, and body mass index (BMI) was calculated. Body weight was measured to the nearest 0.1 kg and height was determined to the nearest 0.1 cm, in subjects wearing underwear and without shoes. Height was measured with a height rod. Bedridden patients were weighed by bed scale and a special weighing set, complete with a digital scale and support spreader bar (Help 2000, Tassinari balance Srl., Bologna, Italy). Bedridden patients’ heights were estimated from ulna length, according to tables provided in the appendix of the MUST screening tool. Information on unintentional weight loss was collected during the interviews with patients and caregivers. Nutritionists used the Global Leadership Initiative on Malnutrition criteria (GLIM) for the assessment of the nutritional status [18]. Two phenotypic criteria—weight loss and low BMI—were used. For etiologic criteria, the inflammation was defined by clinicians, based on the clinical judgement and the assessment of diseases and conditions which may be considered as indicators of inflammation, i.e., chronic organ diseases, and on some laboratory indicators (albumin, prealbumin and CRP). Malnutrition was diagnosed when at least one phenotypic and one etiologic criterion were met, as suggested by GLIM. The malnutrition severity was graded and the distinction between stage 1 (moderate malnutrition) and stage 2 (severe malnutrition) performed, based on phenotypic criteria, using the threshold values proposed by GLIM. Dietitians assessed the texture of diets and protein calorie intake before the visit from food diaries. Wind-food software was used for a detailed nutritional analysis of patient’s intake and needs. Only after a detailed assessment of both the swallowing process and the nutritional status were the personalized nutrition therapies prescribed. Based on the patient’s conditions and needs, the nutritional interventions could imply: (a) the modification of diet form and liquid viscosity, (b) the suggestion for an increase in calorie and/or protein intake, (c) the prescription of oral nutritional supplementation, (d) different combinations of the three or (e) suggestion for AN. The recommended texture of diet was classified according to the International Dysphagia Diet Standardization Initiative (IDDSI) framework, which was adopted by the Clinical Nutrition Unit of INRCA hospital in 2018 [19]. IDDSI consists of a continuum of 8 levels (0–7), where drinks are measured from Level 0 (thin) to 4 (extremely thick), and foods are measured from Levels 3 (liquidized) to 7 (regular). Both oral and written instructions about dysphagia and its management were provided to caregivers who assisted patients at home (family members or informal caregivers who were not sanitary staff). The appropriate calorie–protein intake and the distribution of dietary protein intake across the meals during the day were recommended by the dietitians.

### 2.3. Follow-Up Visits

When it was deemed necessary, the patients were invited to attend a follow-up visit. During the follow-up visit, the patients were weighed, their BMI was calculated, and, when necessary, swallowing was also re-evaluated. The same assessment tools and methods as during the first visit were used. The effectiveness of nutritional therapy in preventing aspiration pneumonia was assessed in all the subjects that attended the follow-up visit. Its effectiveness in the treatment of malnutrition was analyzed in the subjects diagnosed as malnourished at baseline who also attended the follow-up visit, by comparing BMI values registered on the two different occasions.

### 2.4. Data Analyses

The normality of the continuous variables was tested using the Shapiro–Wilk test, and variables were reported as either mean and standard deviation (SD), or median and interquartile range (IQR), on the basis of their distribution. The comparison of the variables between groups was performed according to their distribution by an unpaired Student's *t* test or Mann–Whitney U test. Categorical variables were expressed as absolute frequency and percentage and analyzed by a chi square test. A 2-tailed *p* value of <0.05 was considered significant. Data were analyzed using STATA version 15.1 (StataCorp, College Station, TX, USA).

## 3. Results

In the period from 2018 to 2021, 267 outpatients were assessed for swallowing difficulties and nutritional status at the Clinical Nutrition Unit of INRCA. The most frequent patient-reported symptom of dysphagia was coughing during meals (45.8%), followed by the sensation of a foreign body in the throat (8.3%), oral residue (7.4%), throat clearing (7.2%) and an increased duration of meals (7.2%). Almost 50.0% of subjects reported having two or more symptoms of dysphagia. Most patients (53.9%) reported swallowing difficulties for all consistencies: liquid, solid and mixed. All subjects presented numerous risk factors for dysphagia. As shown in Table 1, the patients were old and very old (mean age 80.5 ± 12.3 years); neurological disease was the most common chronic condition (30.5%). Oral health was poor in more than 50.0% of subjects (poor or no dentition). Almost 90.0% of patients had difficulties communicating, and more than 30.0% were incapable of collaborating with the speech therapist, nutritionist and dietitian during the visit. Subjects mostly had two to seven comorbidities (81.3%) and were using 5 (polypharmacy) to 18 (severe polypharmacy) different drugs (75.0%).

More than 60.0% of drugs used by the assessed subjects were potentially associated with dysphagia [21,22]. Of all the drugs associated with dysphagia, 30.3% were those associated with esophageal damage; 22.7% were drugs that may affect consciousness and alertness, i.e., antipsychotics and neuroleptics; 18.0% were drugs that may cause sleeping and confusion; and 11.0% were drugs that cause xerostomia.

Dysphagia assessment was not possible for eight subjects because of an insufficient level of consciousness and collaboration. Dysphagia was diagnosed in more than 96.0% of subjects who were assessed, according to both GUSS and ASHA. As shown in Table 2, its severity grade was prevalently mild and moderate.

Before the visit, more than 40.0% of subjects were consuming regular or quite regular foods (32.0% Level 7; 11.2% Level 6); more than 65.3% were consuming thin liquids. Only 6.0% of subjects were treated with AN; 1.5% were both artificially and orally fed.

The nutritional screening revealed that 38.2% of subjects were at risk of malnutrition; 22.1% (n = 59) were diagnosed as malnourished (see Table 3. All malnourished subjects also had dysphagia. Malnutrition was severe in half of the cases (49.1%; n = 29) and moderate in the other half (50.9%; n = 30). Malnutrition prevalence was higher in subjects older than 70 years (45.1%) compared to subjects younger than 70 years (23.0%). Almost 10.0% of subjects with dysphagia were obese.

After both dysphagia and nutritional status were assessed, nutritional therapy was prescribed. Texture-modified diets were prescribed or confirmed for 77.4% of subjects who were assessed. Food texture was changed in 33.0% of cases and the texture of liquids in 28.4%. The prevalence of different textures after the visit is presented in Table 4.

In 50.0% of cases, subjects received the suggestion to increase their protein intake; the increase in both protein and calories was suggested in 12.1%.

To increase the calorie and protein intake, oral nutritional supplements (ONS) were mostly recommended. In 6.4% of subjects, AN was prescribed, while in 15.5%, only recommendations about the protein intake in each meal and about the supervision during the meals were given. In 3.5% of cases, only the increase in water intake was necessary.

Follow-up visits were scheduled for 59.9% (n = 160) of subjects; these were actually performed for 63.7% (n = 102) of them. The median number of days between the first and the follow-up visit was 120 (min 30; max 300).

All subjects attending the follow-up visit were following texture-modified diets. Only 1 out of 104 patients (less than 1.0%) reported having experienced aspiration pneumonia since the first visit at the Clinical Nutrition Unit of INRCA. Texture-modified diets successfully treated malnutrition in 13 of the 19 subjects who were diagnosed as malnourished during the first visit and attended the follow-up visit (68.4%). None of them reported episodes of aspiration pneumonia.

As shown in Table 5, 76.9% of the subjects with an improvement of their nutritional status had increased their calorie–protein intake after the first visit; 15.4% had only increased their protein intake. Higher rates of BMI improvement were registered in subjects whose texture of solid foods was modified (30.9% vs. 16.7% in subjects whose texture of solids was not modified), and in younger subjects (81.1 ± 12.1 years vs. 85.7 ± 6.9 years), but the associations were not statistically significant. The number of drugs taken and the proportion of subjects with unintentional weight loss before the first visit were also lower in subjects showing a BMI improvement after the prescribed nutritional therapy (5.92 ± 3.1 vs. 7.3 ± 3.6 and 38.5% and 66.7%, respectively). However, the differences between the two groups were not significant, likely due to the low number of subjects in the follow-up period.

## 4. Discussion

In this study, we reported data of 267 older outpatients whose swallowing difficulties and nutritional status were assessed at the Clinical Nutrition Unit of the geriatric research hospital IRCCS INRCA, Ancona (Italy). Ninety-six percent of subjects were identified as dysphagic; 22.1% were both malnourished and dysphagic. Dysphagia was treated with individualized nutritional therapy: 6.4% of subjects were treated with AN; 77.4% with a texture-modified diet (IDSSI from 6 to 1 for foods and from 4 to 1 for liquids). Among subjects who attended the follow-up visit, less than 1.0% reported having experienced aspiration pneumonia, while the texture-modified diets successfully treated malnutrition in 13 of the 19 subjects who were diagnosed as malnourished during the first visit and attended the follow-up visit (68.4%).

A high prevalence of dysphagia among the patients assessed in this study is not surprising, considering the reported risk factors: old age, the presence of neurological diseases and multiple comorbidities, poor oral health and polypharmacy [23]. In particular, we found that 60.0% of subjects were taking drugs potentially associated with dysphagia, mostly by causing esophageal damage (30.3%); affecting consciousness and alertness (22.7%); causing sleepiness and confusion (18.0%); and causing xerostomia (11.0%). Previous studies have already described the high prevalence of drugs that can cause dysphagia among drug regimens in the elderly. The aforementioned studies argued that drug-induced dysphagia is prevalently caused by xerostomia, while in our study the prevalence of drugs causing dry mouth was almost negligible. Very recently, Wolf and colleagues found that antipsychotics, anti-Parkinson drugs, benzodiazepines and antidepressant medications were associated with a 1.4- to 4.4-fold higher prevalence of OD [21]. Drug therapies represent a modifiable risk factor for dysphagia; therefore, their risk–benefit balance should always be evaluated.

The nutritional assessment that was performed by GLIM criteria revealed that 22.1% of subjects with dysphagia enrolled in our study were malnourished (49.1% severe malnutrition, 50.9% moderate), but almost 68.0% of subjects were not malnourished, and almost 10.0% were even obese. No parallelism between dysphagia and malnutrition was found. Just like Lichenstine and colleagues, we also found that malnutrition was more frequent in dysphagic subjects older than 70 years [24]. In previous studies, different authors have found a prevalence rate of 13% to 55% for malnutrition in subjects with dysphagia [25,26,27,28,29,30,31,32,33]. The differences in the prevalence rates of malnutrition in dysphagia are related to the different settings where the subjects were assessed, but even more to the variety of tools and instruments used to assess nutritional status. Ueshima and colleagues performed the review of the studies assessing the prevalence of malnutrition in adult patients with dysphagia and identified seven nutritional diagnostic criteria used: body mass index (BMI), nutritional screening tool, anthropometric measurements, body composition, dietary assessment, blood biomarkers, and other [25]. They recommended that the GLIM criteria should be used; however, until now, ours is the only study—together with the study of Shimitsku and colleagues—performed using GLIM criteria for malnutrition assessment in subjects with dysphagia [31].

With reference to the nutritional therapies normally adopted to prevent the onset of dysphagia complications, our study confirms that texture modification is a common option (80.6%), while AN is not widely adopted (6.4%). As it was already mentioned, although the texture modification of diets is the most recommended practice, solid evidence on its effects on the prevention of aspiration pneumonia and treatment, as well as the prevention of malnutrition, is lacking. The available evidence derives from observational studies that used very diverse methodologies and numerous classifications of diet textures, which made it impossible to compare their results [34,35,36,37].

In our study, less than 1.0% of subjects that were prescribed an individualized texture-modified diet reported episodes of aspiration pneumonia. Relative to the efficacy of texture-modified diets in treating malnutrition, we found that BMI improved in 68.4% of malnourished subjects following such diets.

Previous studies reported a 1.7-times higher malnutrition risk in LTC residents consuming TMDs, compared with those on standard diets [26,38,39]. Other authors found that only protein and, in particular, calorie-enrichment of texture-modified diets may guarantee weight and BMI improvement [12,26,40,41,42]. Our results also show that BMI improvement probably depends on the adequacy of the energy intake. This result emphasizes the importance of a multidisciplinary approach in the assessment and treatment of subjects with dysphagia. In fact, texture modification without the assessment of patients’ calorie–protein needs and without their adequate provision was not found to be efficient in treating malnutrition in previous studies [12,34,36].

The main strength of our study is that both the assessment of dysphagia and nutritional status and the classification of diet textures were performed using the most recent scales and tools, internationally adopted and approved by scientific societies. Our study also provides important information on drug therapies that represent a potentially modifiable risk factor for dysphagia.

We must also stress some limitations of our study. First, the study had an observational design, and all the statistical analyses performed were descriptive. The limited sample size did not provide sufficient power to test in a multivariable model the effectiveness of an individualized texture-modified diet in preventing aspiration pneumonia and improving BMI. Additionally, the compliance to and the correct application of dietetic prescription was not supervised.

Nevertheless, our results might be helpful in orienting further research, with reference to the main issues requiring investigation, and to the selection of proper tools and scales for nutritional and dysphagia assessment and the classification of diets.

Dysphagia is a topic that requires considerable further research. Clinical trials that offer high-quality evidence on dysphagia and its management are needed. Nevertheless, the use of international standardized instruments and scales within observational studies may also represent a decisive element in overcoming the knowledge gaps that preclude an efficient and evidence-based management of dysphagia. Future studies should also further investigate to what extent the drugs potentially associated with dysphagia are actually responsible for causing it.

## 5. Conclusions

The management of dysphagia and its consequences should never disregard the evaluation of nutritional status; a proper diet for subjects with swallowing difficulties must be characterized by both an adequate consistency and an adequate protein-calorie intake. To generate more quality evidence on the efficacy of texture-modified diets in the management of dysphagia and its complications, it is mandatory that future studies use recently proposed GLIM criteria as the gold standard for assessing nutritional status, and the IDDSI framework, to describe the texture of modified diets.

## Figures and Tables

**Table 1 nutrients-15-01103-t001:** Characteristics of out-hospital patients assessed for swallowing difficulties and nutritional status, and main risk factors for dysphagia–mean ± (SD); absolute and relative frequencies (%).

Gender	161 (60.3) F; 106 (39.7) M
Origin	153 (57.4) Home; 114 (42.6) NH
Age	80.5 ± 12.3
Pluripathologies	217 (81.3)
Main diseases	
Chronic neurological disease	81 (30.5)
Chronic heart disease	27 (10.1)
Chronic respiratory disease	14 (5.3)
Diabetes	15 (5.8)
Chronic kidney disease	7 (2.5)
Cancer	5 (2.0)
All other	117 (43.8)
Polipharmacy	196 (73.5) Yes, 71 (26.5) No
Subjects taking drugs that may cause dysphagia	164 (61.4) Yes, 103 (38.6) No
Oral health	54 (20.1) No teeth, 80 (30.1) Partial, 50 (18.8) Prosthesis, 83 (31.0) With teeth
Collaborative	188 (70.5) Yes, 57 (21.2) Partially, 22 (8.3) No
Trunk control	169 (63.1) Yes, 48 (18.1) Only with support, 50 (18.8) No
Communication deficits	33 (12.4) No, 100 (37.6) Dysphonia, 85 (31.7) Dysarthria, 49 (18.3) Yes, other

**Table 2 nutrients-15-01103-t002:** Dysphagia assessment; GUSS and ASHA absolute and relative frequencies (%).

GUSS	
Slight/No Dysphagia	12 (4.8)
Slight Dysphagia (15–19)	153 (59.0)
Moderate Dysphagia (10–14)	73 (28.1)
Severe Dysphagia (0–9)	21 (8.0)
ASHA	
Level 1	7 (2.7)
Level 2	14 (5.4)
Level 3	18 (6.9)
Level 4	96 (37.2)
Lever 5	79 (30.5)
Level 6	37 (14.2)
Level 7	8 (3.1)

**Table 3 nutrients-15-01103-t003:** Nutritional assessment: malnutrition according to single indicators and according to GLIM criteria absolute and relative frequencies (%).

**Phenotypic criteria**	
BMI	113 (2.3%)
Unintentional weight loss	41 (15.4%)
**Etiologic Criteria**	
Acute disease and chronic disease related	112 (41.9%)
Laboratory measures	68 (25.5%)
**Malnutrition GLIM criteria**	59 (22.1%)

Calorie and/or protein intake was insufficient in more than 50.0% of subjects.

**Table 4 nutrients-15-01103-t004:** Textures of foods and drinks after dysphagia and nutritional status assessment: IDDSI framework; relative frequencies (%).

FOODS		DRINKS	
7	14.0%	4	17.1%
6	18.1%	3	7.3%
5	16.9%	2	11.4%
4	51.0%	1	5.7%
3	0.0%	0	58.5%

**Table 5 nutrients-15-01103-t005:** Descriptive analysis of sociodemographic, functional and clinical parameters, and BMI improvement in dysphagic subjects diagnosed as malnourished during the first visit (relative frequencies, mean values) ± SD); absolute and relative frequencies (%).

	Total	BMI Improvement	No BMI Improvement	*p*
	N = 19 (100.0%)	N = 13 (68.4%)	N = 6 (31.6%)	
Age, mean ± sd	82.6 ± 10.8	81.1 ± 12.1	85.7 ± 6.9	0.411
Gender, n. (%)	4 (21.1) M; 15 (78.9) F	4 (30.8) M; 9 (69.2) F	0 (0.0 M; 6 (100.0) F	0.126
Origin n. (%)	10 (52.6) H, 9 (47.4) NH	7 (53.9) H, 6 (46.1) NH	4 (50.0) H, 3 (50.0) NH	0.876
Pluriphatology n. (%)	19 (100.0)	13 (100.0)	6 (100.0)	-
BMI, mean ± sd	20.1 ± 2.4	19.7 ± 2.5	20.8 ± 2.2	0.365
Malnutrition BMI, n. (%)	16 (84.2) Yes; 3 (15.8) No	11 (84.6) Yes; 2 (15.4) No	5 (83.3) Yes; 1 (16.7) No	0.943
Unintentional LoW n. (%)	9 (47.4) Yes; 10 (52.6) No	5 (38.5) Yes; 8 (61.5) No	4 (66.7) Yes; 2 (33.3) No	0.252
Malnutrition severity, n. (%)	11 (57.9) M, 8 (42.1) S	6 (46.2) M; 7 (53.8) S	5 (83.3) M; 1 (16.7) S	0.127
N. drugs, mean ± sd	6.4 ± 3.2	5.9 ± 3.1	7.3 ± 3.6	0.398
N. dysphagia drugs, mean ± sd	0.8 ± 0.8	0.8 ± 0.8	0.7 ± 0.8	0.693
GUSS, mean ± sd	15.0 ± 3.2	15.1 ± 3.7	14.8 ± 1.9	0.874
ASHA NOMS, mean ± sd	4.2 ± 0.9	4.2 ± 1.1	4.0 ± 0.6	0.639
Modification of solids n. (%)	5 (26.3) Yes; 14 (73.7) No	4 (30.87) Yes; 9 (69.2) No	1 (16.7) Yes; 5 (83.3) No	0.516
Modification of liquids n. (%)	4 (21.1) Yes; 15 (78.9) No	2 (15.4) Yes; 17 (84.6) No	2 (33.3) Yes; 4 (66.7) No	0.372
Intake increase n. (%)	13 (68.4)	10 (76.9)	3 (50.0)	0.251
Calorie–protein	5 (26.3)	2 (15.4)	3 (50.0)	
Protein None	1 (5.3)	1 (7.7)	0 (0.0)

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
