# Peer review of "Multidisciplinary Assessment and Individualized Nutritional Management of Dysphagia in Older Outpatients"

_nutrients, 2023, doi:10.3390/nu15051103_

Round 1
Reviewer 1 Report
1. What is the main question addressed by the research? Yo describe the characteristics of older outpatients, whose swallowing difficulties and nutritional status were assessed in a geriatric research hospital, by tools and scales that the scientific community has recently proposed. 2. Do you consider the topic original or relevant in the field? Does it address a specific gap in the field? Yes. Dysfagia is a huge topic that need additional research. 3. What does it add to the subject area compared with other published material? Based on this finding, the main strength of our study is that both the assessment of dysphagia and nutritional status and the classification of diet textures was performed using the most recent scales and tools, internationally adopted and approved by scientific societies. The study provides also important information on drug therapies that represent a potentially modi fiable risk factor for dysphagia which could be managed in order to prevent the onset of swallowing disorders. 4. What specific improvements should the authors consider regarding the methodology? What further controls should be considered? In my opinion the methodology used was the correct one. 5. Are the conclusions consistent with the evidence and arguments presented and do they address the main question posed? The seccion "Conclusion" should be added to the paper. 6. Are the references appropriate? In my opinion, yes! 7. Please include any additional comments on the tables and figures.Tables and figures are adequate, in terms of information and graphics. 8. Other comments: Abstract should be organized in seccion (introduction, methods, results, conclusion).
Author Response
Thank you for your observations and suggestions.
As you suggested, we added the section “Conclusion” to our manuscript and highlighted it in red.
We also appreciated your suggestion to structure better our abstract. We added to it all sections (Introduction, methods, results, and conclusion) and highlighted it in red.
As you can notice, following the suggestions of another reviewer, we added to our Abstract some information about statistical procedures and had to reorganize the text in the section “Methods” in the Abstract.
We also added some references and moved other references in the text, you can find all changes highlighted in the text.
Reviewer 2 Report
Multidisciplinary assessment and individualized nutritional management of dysphagia in older outpatients – is an interesting paper that discusses the assessment of individual nutritional management of older patients suffering from dysphagia.
Some recommendations and questions are provided.
Abstract – should include the statistical procedures done on what data collected (one simple sentence will do) this would let readers better understand what was done, and how/where the results/conclusion were drawn from.
In the introduction – should include some information for the GLIM criteria – when and why the sudden need to use GLIM criteria. Just to connect to the research objective and significant of the study.
Method/procedure are clearly described and adequate.
Data analysis – was wondering did you try to use the regression? Or try to control the effects of the background demographics variables? Nonetheless, analyses seem fine.
Discussions – just a suggestion – would recommend to add a conclusion section – the information is within the discussions and could just be expanded.
Would also suggest to provide clear research objectives (more specific), since there are various results /tables that describes various information and assessments.
Should also expand the implications of the study, what now?
Author Response
Thank you for your recommendations and suggestions.
As you suggested, in order to improve the comprehensibility of the manuscript we added a sentence about statistical procedures in the Abstract. We highlighted the text that was added in red.
As you suggested, we explained in the Introduction the reasons for the use of GLIM criteria and IDDSI scale. We highlighted the text that was added in red.
Relatively to you question about the statistical procedures (“Did you try to use regression? Or try to control the effects of the background demographics variables? ), we actually have tried to run some simple logistic regressions on BMI Improvement to verify if at least some of the demographic or independent variables may have same statistically significant relationship. Unfortunately the sample is to small to obtain any significant estimation and this can be inferred also from table 5, where no comparison lead to a p<.05.
Thank you for your suggestion to add conclusion section in Discussion. We followed your recommendation and highlighted the text in red.
Thank you for your observation about the necessity to explain better the research objectives. We added some explanations in the Introduction and highlighted it in red.
As you suggested, we evidenced better in the Discussion the implications of the study and what should be done in the future. We believe that our study draws the lines on how to proceed in the collection of evidence that could be useful for the management of dysphagia. The text is highlighted in red.
Due to some changes in the text, we moved some references and added others. You can find all changes highlighted in the text.